# Recent Advances in CRP Biosensor Based on Electrical, Electrochemical and Optical Methods

**DOI:** 10.3390/s21093024

**Published:** 2021-04-26

**Authors:** Seungwoo Noh, Jinmyeong Kim, Gahyeon Kim, Chulhwan Park, Hongje Jang, Minho Lee, Taek Lee

**Affiliations:** 1Department of Chemical Engineering, Kwangwoon University, Seoul 01897, Korea; nsw26510@naver.com (S.N.); wls629@icloud.com (J.K.); 1497rg@hanmail.net (G.K.); chpark@kw.ac.kr (C.P.); 2Department of Chemistry, Kwangwoon University, 20 Kwangwoon-Ro, Nowon-Gu, Seoul 01897, Korea; hjang@kw.ac.kr; 3School of Integrative Engineering, Chung-Ang University, Seoul 06974, Korea

**Keywords:** CRP, biomarker, biosensor, biomaterials

## Abstract

C-reactive protein (CRP) is an acute-phase reactive protein that appears in the bloodstream in response to inflammatory cytokines such as interleukin-6 produced by adipocytes and macrophages during the acute phase of the inflammatory/infectious process. CRP measurement is widely used as a representative acute and chronic inflammatory disease marker. With the development of diagnostic techniques measuring CRP more precisely than before, CRP is being used not only as a traditional biomarker but also as a biomarker for various diseases. The existing commercialized CRP assays are dominated by enzyme-linked immunosorbent assay (ELISA). ELISA has high selectivity and sensitivity, but its limitations include requiring complex analytic processes, long analysis times, and professional manpower. To overcome these problems, nanobiotechnology is able to provide alternative diagnostic tools. By introducing the nanobio hybrid material to the CRP biosensors, CRP can be measured more quickly and accurately, and highly sensitive biosensors can be used as portable devices. In this review, we discuss the recent advancements in electrochemical, electricity, and spectroscopy-based CRP biosensors composed of biomaterial and nanomaterial hybrids.

## 1. Introduction

With continuous advances in science and technology, the lives of people worldwide have been enriched and the average lifespan extended. Nevertheless, humans grow old and this process leads to several diseases such as myocardial infarction [1,2], hypertension [3,4], and immune diseases [5,6]. Biomarker detection is an efficient method for the rapid diagnosis of disease. C-reactive protein (CRP) is a biomarker associated with inflammatory processes, cardiovascular diseases.

Inflammation is a defense mechanism of the body in response to external stimuli from damage, infection, or disease. [7,8,9]. Wound healing or tissue repair begins during an inflammatory response, stimulates angiogenesis, and promotes the production of neutrophils, macrophages, and lymphocytes that protect the body from foreign antigens. During the inflammatory response, damaged tissue cells promote the production of creatinine, tumor necrosis factor (TNF-α), interleukin 6 and C-reactive protein (CRP) [10,11,12]. CRP is one of the best-known acute phase proteins and is comprised of five subunits of the same polypeptides [13]. Acute-phase proteins exist in the blood in physiological conditions, and their production is increased or decreased rapidly due to various diseases or threats such as injury, infection, acute myocardial infarction, and cancer [14]. Blood CRP levels can increase up to 1000 times after stimulation, such as acute inflammation, with a half-life of 19 h, before quickly returning to normal blood levels [15]. Elevated levels of CRP are used as an important indicator for the possibility of cardiovascular disease [16], and the high CRP concentration due to chronic inflammation may affect the cancer development process [17].

CRP is synthesized in the liver and circulates in the body via plasma. Under normal conditions, the plasma concentration of CRP is generally less than 2.0 mg/L [18]; excess CRP concentrations above 2.0 mg/L can therefore be used to suspect pathologies such as cardiovascular disease, infection, and inflammation [19].

The CRP level test is performed by collecting venous blood and using an enzyme-linked immunosorbent assay (ELISA) [20,21] or an immunoassay applied to ELISA, providing accurate information about the CRP level in the blood. however, the test itself is time-consuming, requires complex detection steps and professional manpower. Therefore, in order to overcome these limitations, various methods have been proposed to detect CRP using other methods such as fluorescence [22], surface plasmon resonance (SPR) [23], surface-enhanced Raman spectroscopy (SERS) [24], and electrochemistry (EC) [25]. However, it still had several limitations, such as requiring a labeling process or a high detection limit. To overcome these limitations, researchers attempted to introduce nanomaterials into biosensors. The development of nanotechnology has greatly influenced the development of biosensors [26,27]. Nanomaterials offer novel opportunities for the construction of biosensors and the development of new bioassays [28,29]. Nanomaterials are generally used for enhancing electrochemical reactions [30], labeling biomaterials [31], or increasing binding events [32]. To shed light on the current progress of the CRP detection platform, we review the recent advances in CRP detection systems divided into three sections, including the latest research trends based on EC, SPR, SERS, etc.

## 2. Field-Effect Transistor-Based Biosensors

The biosensor using the electrical sensing method is the preferred method for biosensors now because the measurement time is relatively short compared to other measurement methods and does not require expensive measurement equipment [33]. In particular, the field-effect transistor (FET) has been the subject of substantial research and development in recent years, and it has gained popularity in numerous applications used on a daily basis. Biosensors using FETs are currently under research due to their advantages of miniaturization, mass production, and inexpensive production processes [34,35].

FETs are composed as shown in Figure 1A. The principle underlying FET technology is that upon the application of a voltage to the gate (G), the minority carriers in the body are gathered toward the gate to form a channel, and the source (S) and drain (D) are connected. At this time, when a voltage is applied to the drain, a current flows from the drain to the source, through the channel [36]. P-type and N-type FETs differ depending only on whether this minority carrier is an electron or a hole, but the main principle underlying each of these types remains the same.

The FET biosensor is a method that measures the current change of the transistor through the reaction between the surface of the thin film transistor and the target; the current change is sensed through the change in the threshold voltage and conductivity of the gate voltage [37,38].

In particular, research by the Lee group has clearly established that the sensing method is based on the change in threshold voltage [37]. Their study proposed a sensor system composed of heterogeneous AlGaN/GaN layers as shown in Figure 1C. Ni/Au (20/50 nm) was deposited as a gate metal, and a CRP antibody was immobilized thereon. One of the two transistor structures was used to detect CRP, and the other was used as a reference to eliminate the noise factor. The two were almost electrically identical, and the gate length and width of the fabricated sensor were 20 μm and 100 μm, respectively. The FET operated in a constant drain–source current (IDS = 2.5 mA) and drain-source voltage (VDS = 1.5 V) mode. The immobilization and subsequent coupling between 11-mercaptoundecanoic acid (11-MUA), CRP-antibody and CRP in Figure 1D could be confirmed through the threshold voltage (Vth) movement direction. A constant current method was used to determine Vth. First, 11-MUA was fixed to the Au gate, and Vth shifted to the right due to the net negative charge of the thiol group of 11-MUA. When exposed to CRP-antibody, Vth shifted to the left due to the positive charge of the amine group of CRP-antibody and, finally, when specific binding between CRP and CRP antibody occurred, Vth shifted to the right due to the negative charge of CRP. As a result, Vth shifted from −2.25 V to −1.64 V. This was nonlinearly proportional to the CRP concentration from 0.01 ng/mL to 1000 ng/mL, indicating the sensor system could be effectively applied to a portable biomolecule detection system.

Another study demonstrated that CRP sensing could be achieved through the change of drain current in a FET-based biosensor [38]. As shown in Figure 1E, an AlGaN/GaN heterolayer FET system was constructed. When the CRP antibody was fixed to the gate metal and specific binding occurred between CRP and the CRP antibodies, the piezoelectric-induced carrier density in the FET channel was changed, resulting in a change in the drain current. The drain current of the sensor was driven by a current source (IS2) and VDS was fixed by a current source (IS1) and a resistor (RSET). The change in surface potential led to a change in Vth of the FET biosensor. Figure 1F shows the ID-VDS characteristics of the biosensor as a result of the specific combination of the self-assembly monolayer (SAM) composed of 11-MUA, an anti-CRP antibody, and CRP.

The drain current of the SAM decreased by 2.98 mA at VDS = 8 V. In this case, a negative potential was introduced to the gate surface due to the negative charge of the MUA thiol group. After the SAM was formed, the CRP antibody was immobilized, and the positive charge of the CRP antibody amine group attracts electrons from the channel. Therefore, as the CRP antibody binds to the SAM, the drain current of the sensor increases by 1.68 mA. Finally, when CRP and CRP antibodies specifically bind, the drain current is reduced due to the negative charge of CRP. Using the proposed sensing system. It was shown that CRP can be detected at various concentrations from 10 ng/mL to 1000 ng/mL. Table 1 shows biosensing techniques using electrical means of detection that can be used to detect probes and CRP representing LOD.

In summary, FET-based biosensors can quantitatively analyze antigens using the gate surface potential and current that change when an antigen–antibody reaction occurs with the target by immobilizing a target recognition material on the gate. As the FET system has the ability to amplify signals by the fundamental structure along with such current change, it can be ultrasensitive detection, and it is possible to detect at high speed without the need for a separate transducer. However, there is a limitation that the analyte must be charged, and the distance between the gate surface and the surface material must be within the Debye length to be analyzed [39]. Therefore, it is necessary to look at other analysis methods to overcome these limitations.

## 3. Electrochemical Biosensor

Over the last two decades, electrochemical biosensor platforms have been used as precise and sophisticated analytical methods, as their advantages include ease of handling, cost-effectiveness, high sensitivity, high selectivity, fast response time and easy operation [40,41]. Many electrochemical analysis techniques such as conventional electrochemical methods [42,43], photoelectrochemical methods [44,45], and electrochemiluminescent methods [46,47] are used for chemical and biological analysis across a wide range of electrochemical reactions [48,49] on electrodes and electrolyte interfaces. Among them, electrochemical sensors are attractive due to their high specificity, sensitivity, simplicity, low energy consumption, and a wide range of applications; as such, their development has rapidly increased in recent years [50,51]. In addition, electrochemical sensors and biosensor platforms, which combine multifunctional nanomaterials [52] and analytical technologies [53], can be applied extensively in clinical and medical diagnostics [54], as well as health monitoring [55]. The potential applications of these platforms are greatly facilitated by their advantageous features such as miniaturization, simple production processes, low cost, and fast detection capabilities [56].

Electrochemistry is the study of electricity generated by the transfer of electrons between substances in a chemical reaction [57]. The electrochemical sensor detects an antigen using current or electron transfer resistance generated by redox species. Cyclic Voltammetry (CV), Differential Pulse Voltammetry (DPV), Square Wave Voltammetry (SWV) and Electrochemical Impedance Spectrum (EIS) are commonly used as electrochemical techniques. CV, DPV, and SWV are voltammetry techniques [58,59]. These techniques are a method of recording current by applying voltage in the form of pulses. Each method is classified according to the pulse waveform. EIS is an effective electrochemical impedance spectroscopy method that measures the impedance value of the electrode surface during the frequency change process. The antigen reacts with the bioprobe to detect analytes through changing signal strength. Figure 2A shows a diagram of an electrochemical sensor. This section introduces sensors based on electrochemistry.

Nanomaterials can be used as an electrochemical sensing material for biomedical and biological applications, due to their excellent surface charge, physicochemical properties, surface advantage, shape and inclusion of a catalyst. The nanomaterial’s crystal structure, surface roughness, and chemical properties help detect antigens by controlling the electron transport mechanism [57]. Nanomaterials (noble metal nanomaterials, metal oxide nanomaterials, carbon nanomaterials, polymer nanomaterials and nucleic acid-based bio-nanomaterials), described above, have been used to enhance electrochemical sensors and biosensor platforms [60,61]. Recently, the Tang group published a study on improving the signal strength of electrochemical using organic–inorganic hybrid nanoflowers [62]. The nanoflowers used in the Tang group are BSA-antibody-copper phosphoric acid hybrid nanoflowers (BSA-Ab2-Cu3(PO4)2), which act as signal enhancers for enzyme-free electrochemical immunoassay devices. The three-dimensional porous nanoflower form, with its high specific surface area, greatly increases the sensitivity of the manufactured sensor; BSA is loaded to block non-specific sites, many antibodies can be used, and the hybrid nanoflower also provides a large amount of phosphate anions which react with molybdophosphate to produce an electrochemical signal for detection. Figure 2B shows the synthesis process of BSA-Ab2-Cu3(PO4)2hybrid nanoflowers and a schematic image of the technology. The characteristics of the completed electrochemical immunosensor are shown in Figure 2C. As shown in Figure 2C, a noticeable redox peak appears at 0.34 V and 0.14 V using the Cyclic Voltammetry (CV) technique [63,64], which is confirmed to result from electron transfer of Mo in molybdophosphate [65,66]. As no redox peak was found in the red and blue lines, it was confirmed that molybdophosphate forms redox currents due to the electron transfer of Mo. In addition, its redox activity was confirmed by square wave voltammetry (SWV) [67,68]. A sample without CRP and a sample with CRP were compared using the fabricated electrochemical immunosensor. Thereafter, when molybdate was loaded, a remarkably higher current peak appeared in the sample with CRP than in the sample without CRP. Through this, the specific binding between the immobilized antibody of the hybrid nanoflower and the CRP immobilized on the electrode was confirmed. Figure 2D shows the electrochemical impedance spectrum (EIS) [69,70], which is composed of a linear part at low frequency and a semicircle part at high frequency. This was conducted to confirm the assembly process of the electrochemical immunosensor. The diameter of the semicircle represents the impedance of the electrode, and the impedance of each electrode is observed to change noticeably during the electrochemical process. Compared with bare GCE (curve a), the electron transfer resistance gradually increased after immobilizing polydopamine nanospheres (PDANS), Ab1, BSA, CRP, and nanoflowers (curve b~f) continuously. The process of this electrochemical immunosensor is successful and effective because proteins increase impedance by interfering with electron transfer. To understand the analytic performance of the immunosensor, the range of CRP at different concentrations was measured with SWV technology. As the concentration of CRP increased, the SWV peak current of the immunosensor increased at 0.14 V and 0.34 V, respectively. This was found to occur because, as more CRP was immobilized on the electrode, a larger amount of nanoflowers were formed on the electrode to precipitate molybdophosphate, resulting in a higher redox peak current. Calibration curve for immunoassay of the peak current signal measured at SWV 0.14 V, and LOD was measured as 1.26 pg/mL.

The Dong group also developed an electrochemical immune response sensor, based on ZnO/porous carbon matrix (ZnO/MPC) as a new nanomaterial for analysis of CRP protein [71]. Their electrode, with an immobilized ZnO/MPC [72] and ionic liquid (IL) composite membrane, displayed excellent conductivity and biocompatibility for the ultra-sensitive detection of CRP. Figure 2E shows a schematic image of the study. For quantitative determination of various CRP concentrations (0.01–1000 ng/mL), the performance of the ZnO/MPC-based immunosensor was evaluated by differential pulse voltammetry (DPV) [73,74]. As shown in Figure 2F, as the concentration of CRP in this immunosensor increases, the current peak gradually decreases, indicating that the CRP complex interferes with the electron transfer occurring on the electrode surface. The calibration curve appeared linearly (R = 0.0998) over a wide range of 0.01 to 1000 ng/mL, and the detection limit was determined to be 5.0 pg/mL.

Nucleic acid aptamers are short oligonucleotides developed with high specificity and affinity to interact with the target substances [75]. Aptamers are chemically modified and relatively easily prepared, selected, and used for various biomedical applications such as targeted therapy [76]. Wang group introduced RNA aptamer-based electrochemical aptasensor [77] for C-reactive protein detection using functionalized silica microspheres as immunoprobes [78]. The nanomaterials used in this study functioned gold nanoparticles (Au NPs) in silica microspermes (Si MSs) [79] with uniform shape, providing a wide surface area so that signal molecules (Zincs) and antibodies (Ab) can be immobilized well. Figure 2G shows the synthesis process of nanomaterials and the production process of CRP aptasensor. RNA aptamers designed to specifically recognize CRP are immobilized on the electrode surface modified with Au NPs through gold-sulfur affinity. In this immunosensor, a sandwich structure of RNA aptamer-CRP-nanomaterial is formed, and electrochemical signals are recorded using the SWV technique. Figure 2H evaluated the performance of aptasensor using the SW technique. As the CRP concentration increases, the signal intensity of Zn^2+^ increases, and an increase in the current peak can be observed. The detection limit accordingly was 0.0017 ng/mL, and a linear calibration curve was obtained in the range of 1 to 125 ng/mL (Figure 2I).

Lee group fabricated a CRP biosensor comprised of multi-fuctional DNA (MF-DNA)/a porous rhodium nanoparticle (pRhNP) heterolayers on Micro-gap using an electrochemical method [80]. To construct a CRP bioprobe, a DNA 4way-junction (4WJ) was introduced. DNA 4WJ consists of a recognition region for detecting CRP, a region generating an electrochemical signal amplification, and an anchoring region (Thiol group), respectively. The electrochemical biosensor was constructed by immobilizing pRhNP on Micro-gap and then prepared MF-DNA/AgNO_3_ was immobilized onto pRhNP. The binding between the target and the bioprobe was confirmed through CV, EIS measurement. Quantitative evaluation of sensitivity was performed using serial dilutions of CRP (1 pM–100 nM) in 10 mM HEPES and 5 mM [Fe(CN)_6_]^3−/4−^. As a result, it appeared linear in the CRP concentration range of 1 pM–100 nM, and the LOD was determined to be 0.349pM. In the same way, linearity was shown in the CRP concentration range of 1 pM–100 nM in 20% diluted human serum, and the LOD is 3.55 fM, which shows that the sensor can detect the humoral response at a very low detection limit. Table 2 details the electrochemical biosensing techniques for CRP detection, indicating the probe and LOD.

In summary, the electrochemical biosensor quantitatively analyzes the antigen using the current or electron transfer resistance generated on the surface by redox species when an antigen-antibody reaction occurs. However, the electrochemical method has a limitation that the electrical redox ability must be different from that of other interfering substances. There is also a disadvantage of high LOD. It is necessary to explore other methods of analysis to overcome these limitations.

## 4. Optical-Based Biosensor

Optical-based biosensors are a method that converts the specific signals in optical properties such as absorption, reflection, and emission of light. In particular, the optical properties of nanoparticles have been using as the basis for signal amplification. The advantages of optical biosensing are high specificity, fast detection, high sensitivity, and real-time monitoring. Optical biosensors utilize an electric field, which is an optical field interaction with an analyte. The basic mechanism is when polarized light hits a metal film under total internal reflection, and an extinction wave is generated. When total internal reflections occur in a prism covered with a metal film, the reflected photons generate an electric field on the opposite side of the prism–metal interface. When photons are sufficiently absorbed and are interact with free electrons (resonances) in the metal membrane, it is converted into surface plasmons that vibrate free electrons. As the composition of the medium changes, the momentum of the plasmon changes, causing resonance and the angle of incident light changes. This change in refraction angle is measured by surface plasmon resonance (SPR). For example, the light of a specific wavelength is absorbed or reflected by a metal surface. One of the interacting substances is fixed to the sensor surface, the other passes through a free and fixed interaction in the solution. The binding and dissociation of two interacting substances are measured in real-time according to the change in refraction [81,82].

Optical biometrics methods can be divided broadly into methods using fluorescence [83,84], colorimetry [85,86], Raman spectroscopy [87,88] and surface plasmon resonance (SPR) methods [89,90]. Without a label or in a label-based method. Label-free detection involves the interaction of an analyte with a transducer for signal detection. On the other hand, label-based detections are the method of labeling analytes and measuring optical signals by colorimetric, fluorescence, or luminescence.

SPR is a phenomenon wherein polarization occurs when light energy is absorbed by the surface of a conductive material. Electrons are subsequently formed by increasing the intensity of an electric field by polarization. This phenomenon is used to monitor the process of binding between proteins and antigens. Its main advantage is that SPR can detect target substances more rapidly compared to conventional ELISA assays and, instead of detecting antibodies indirectly, the antibody binding process can be visualized in real-time [91,92]. Moreover, SPR constitutes the only label-free biosensor technology among the currently established optical sensors [93].

Aray et al. proposed a plastic fiber optic biosensor to fabricate an SPR-based immune sensor using antigen–antibody assays for CPR detection [94]. The SPR biosensor used a solution of 11-MUA in H_2_O/ethanol (10% ethanol) to functionalize the sensor’s gold surfaces, and then contact the thiol solution with the gold layer. Before the -COOH group was activated by a crosslinking chemical reaction after fixing the CRP antibody, BSA was adsorbed on the surface to block the activated carboxyl group and prevent non-specific binding. The CRP protein was then measured using the concentration of human serum as the target. Used this sensor, a linear detection range of 0.009 mg/L CRP was found in the concentration range of 0.006–70 mg/L in human serum and was detected in real complex media and serum. The SPR platform is a high-sensitivity, real-time, label-free, portable, and low-cost biosensor that meets the performance requirements of clinical applications.

Wu et al. detected CRP using a DNA aptamer obtained using a microfluidic chip. First, a surface plasmon resonance biosensor enhanced with Au nanoparticle constructed using a DNA aptamer obtained from a fluid chip and then introduced to detect CRP at concentrations ranging from 10 pM to 100 nM in diluted human serum. This study has shown practical applicability in the diagnosis of CRP disease in human serum [95]. Other sensors detected by SPR include Meyer’s antigen–antibody analysis for CPR detection [96].

Additionally, local surface plasmon resonance (LSPR) has been applied to these sensors [97,98]; in LSPR, vibration is generated on the surface of metal nanoparticles with local surfaces on the SPR sensor [99,100]. Figure 3A shows the basic principle of LSPR, which can detect biological interactions on the nanoscale due to the bonding between the electromagnetic field (EM) and spatially constrained free electrons. LSPR is a non-propagating plasmonic excitation and can be resonantly excited around metal nanoparticles or nanoholes in thin metal films. The operational detection principle relies on LSPR spectral shifts due to local genomic medium changes due to biological binding events. Since the resonance conditions of LSPR are quantified by the motion of electrons, their optical sensing properties are highly dependent on the geometry of the metal nanostructures. To achieve LSPR, these nanostructures can resonate with the occurrence of an EM field at a specific wavelength, leading to a strong near field [101].

A variety of nanoparticles are used to generate the LSPR phenomenon and increase detection sensitivity. Generally, the most widely used particles for this purpose are gold, silver, and aluminum; in particular, gold has high biocompatibility and stability [102,103].

The binding event of the bioprobe and the virus on the surface of such nanoparticles can be confirmed by changes in the intensity and length of the wavelength, and LSPR can be confirmed using this difference. LSPR-based biosensors are excellent for use field detection due to their structural simplicity, easy operability, and portability [104].

Yeom et al. developed a sensor that improves the sensitivity of the CPR biosensor [105]. A CRP antibody on gold nanoparticle (GNP)-labeled optical interference with the LSPR produce. After the anodicaluminum oxide (AAO) chip constructs, the gold nanoparticles were combined with nickel. A thiol group was subsequently formed on the gold surface using 11-MUA acid containing a carboxyl group, and a CRP antibody was bound. Additionally, EDC and NHS were used to assist in the binding of CRP antibodies to gold.

Figure 3B briefly shows the structure of the AAO chip. A proposed LSPR biosensor showed a high sensitivity of 868 nm/RIU to changes in the surface refractive index of gold nanoporous membranes and detects 1 fg/mL of CRP antigen shown in Figure 3C. Moreover, a sandwich immunoassay can be applied using a secondary antibody labeled with GNP to detect up to 100 ag/mL (Figure 3D). This biosensor had proven that the LSPR-based biosensor is a sensitive sensor. The result can apply to other biosensor systems and bioimaging systems. Additionally, a sensor has been established to measure CRP using poly(2-methacryloyloxyethylphosphorylcholine)-grafted gold nanoparticles prepared through surface-initiated atom transfer radical polymerization by LSPR [106].

The colorimetric sensor checks the color change caused by LSPR caused by metal NPs. One of the most properties of metal NPs such as Au and Ag is that LSPR occurs in the metal NP, which represents the collective vibration of conduction electrons when the frequency matches the frequency of the incident electromagnetic radiation. The optical plasmon properties of metallic NPs are highly dependent on the interparticle distance between NP pairs, small or large aggregates of metallic NPs, compared to individual NPs and well-spaced NPs. As the interparticle distance decreases, a strong overlap occurs between the plasmon fields of nearby particles, increasing the intensity and causing a redshift in the LSPR band, making it easier to observe the change in the color solution. Au and Ag NPs provide excellent LSPR properties that exhibit strong, well-defined colors, and the color change between individual NPs and well-spaced NPs can easily be visualized or confirmed by UV-visible spectrometry compared to aggregated NPs [107].

In a study by António, CRP was detected quickly and using an AuNP-aptasensor [108]. The overall schematic diagram of the CRP detection method is the same as in Figure 3E. After preparing citrate-covered gold nanoparticles (AuNPs), an aptamer solution adds the AuNPs colloid, then CRP solutions add incubating the resulting mixture in an orbital shaker incubator. Then, through UV-vis spectrophotometry analysis, the aggregation rate was confirmed, and the absorbance according to the aggregation state of Au NPs was determined. Figure 3F shows the measurement of absorbance at each stage of the sample. The CRP concentration in the sample is the aggregation ratio (A670 nm/A520 nm). Linearity of 0.889–20.7 μg/mL was confirmed in Figure 3G, the LOD was 1.2 μg/mL, which was similar to the typical clinical cutoff concentration for a high-sensitivity CRP assay (1 μg/mL).

Byeon et al. developed a simple, rapid colorimetric immunoassay system for CRP detection based on gold nanoparticles. After preparing AuNPs with CRP antibodies, aggregation of AuNPs according to the amount of CRP appears in a unique pentameric structure. Aggregation by antibody-binding AuNPs and CRPs resulted in a maximum wavelength change in the UV/Vis spectrum. The maximum shift in absorption maxima occurs when the concentration of the CRP antigen is 100 ng/L. it for qualitative analysis of CRP in serum samples applicability of the new homogeneous assay system. Combined observations showed that the method using CRP antibody-conjugated AuNPs is quick and straightforward, and therefore promising for applications to POC diagnosis [109].

Raman spectroscopy sensors are used as molecular identification tools. that measure the vibration spectrum of molecules, enabling qualitative and quantitative analysis of molecules [110,111,112]. Raman scattering mainly relies on the energy loss (Stokes) or gain (Anti-Stokes) of inelastic scattering photons due to molecular vibration events and reflects molecular structure information to enable in-situ and real-time detection.

Figure 3H shows the basic principle of Surface-enhanced Raman spectroscopy (SERS). First, when the chemical target material is close to a particular metal nanosurface, the wavelength of the incident light produces surface plasmons (electromagnetic enhancement) on the metal surface. Then, surface plasmon is the principle that Raman signal is amplified through interaction (chemical enhancement) with the analytical substance. The SERS is a subset of Raman scattering and provides a million-fold improvement through plasmonic nanostructures, lowering detection sensitivity at the single-molecule level. The enhancement of SERS enhancement divided into two distinct mechanisms: electromagnetic enhancement and chemical enhancement. The key to electromagnetic enhancement is to combine the incoming light with LSPR of plasmonic nanostructures, effectively focusing the electromagnetic field on the secondary electric field. The enhanced electromagnetic field amplifies the SERS intensity as the molecule approaches the plasmon nanostructure (distance less than 10nm). Electromagnetic strengthening also occurs when molecules bind directly to the surface of the plasmonic surface, creating a charge transfer structure. Chemical strengthening is related to the intrinsic chemical structure of the analyte molecule and its affinity for the surface, but the contribution of chemical strengthening is usually less than that of electromagnetic strengthening [113].

The SERS can detect non-destructive analytes that are not labeled due to spectral results, such as fingerprints of biochemical molecules, and detecting individual components of multi-component substances [114,115].

In Liu’s study, CRP was detected using Fe_3_O_4_ @ Au SERS tag-based lateral flow analysis [116]. The EDC and NHS activate Fe_3_O_4_ @ Au MNP. The activated product was separated with a magnet, then incubated with the antibody with agitation in PBST. BSA blocks carboxyl groups. Subsequently, LFA strips were assembled, and CRP detection antibody and polyclonal goat anti-mouse IgG were sprayed onto the NC membrane in two lines. The LOD of CRP measured in the Fe_3_O_4_ @ Au SERS tag-based lateral flow analysis was determined to be as low as 0.01 ng/mL.

Moreover, the SERS platform using porous magnetic Ni@C nanospheres and CaCO_3_ microcapsules has been established [117]. The SPR biosensing techniques for CRP detection, which showed the probe and LOD, are presented in Table 3.

## 5. Future Perspective

In modern society, the rate of chronic disease is rapidly increasing worldwide due to age-related risk factors and lifestyle factors. CRP is an acute inflammatory protein that is increased up to 1000-fold in inflammation or infection sites and is being studied as a biomarker of inflammation. CRP is synthesized in the liver and circulated in plasma, and health status can be identified through a change in CRP concentration. Therefore, fast and accurate detection of CRP concentration is an important area of diagnostics. The biosensor reacts with targets to be measured, such as antigen–antibody interactions, to quickly and accurately analyze substances. In other words, the biosensor has the advantage of excellent measurement simplicity, speed and sensitivity. As discussed here, electrical, electrochemical, and SPR biosensors are strong candidates for CRP diagnostics. However, each biosensor technique has its own distinct disadvantages. New research needs to be conducted to compensate for limitations such as the difficulty of miniaturization, high process cost, and high LOD.

Nanomaterials applied in electrical, electrochemical and SPR biosensors are used as materials that enhance the detection of target materials. The advantages of nanomaterials are as follows: (1) nanomaterials can be applied with various advantages such as physicochemical properties, surface area advantages, and catalysts; (2) nanomaterials can be modified easily through chemical methods, and applied to detection technology; (3) novel forms of technology can be created by converging existing technologies in nanomaterials research. These advantages indicate the possibility of fast and accurate nanomaterial-based biosensors that could be commercialized and used in the field. However, some concerns exist regarding the impact of nanomaterials on the environment, ethical and social aspects of nanomaterial development, and human safety issues. Additionally, the possibility of the toxicity of nanomaterials, their penetration into the human body, and environmental pollution warrant further research to inform safe and effective development of these materials. Nanomaterial-based biosensors will benefit from further research and development, as they are shown to be effective for CRP detection, and are an attractive tool for the early diagnosis of diseases such as cardiovascular disease. The next challenge of nanomaterial-based biosensors will be a rapid measurement, field-ready module construction with a smartphone-integrated type, and the miniaturization of biosensors for the mass production of not only CRP detection but also general CVD diseases.

## Figures and Tables

**Figure 1 sensors-21-03024-f001:**
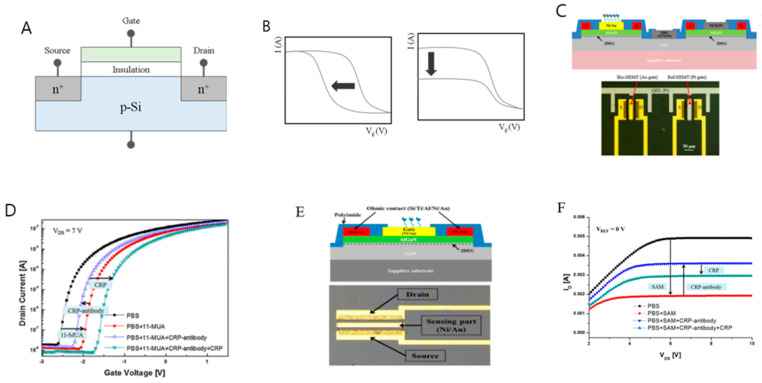
(**A**) Field-effect transistor (FET) structure. (**B**) Example of detection using change in threshold voltage at drain and current change at drain. (**C**) FET-based sensor system composed of heterogeneous AlGaN/GaN layers. (**D**) The state of the gate viewed through the change in the threshold voltage at the drain. (**E**) Another FET-based sensor system composed of heterogeneous AlGaN/GaN layers. (**F**) Gate state observed through current change in drain (Reproduced with permission from [37,38], published by Elsevier 2016, the figure follows the term of use under a Creative Commons Attributions License).

**Figure 2 sensors-21-03024-f002:**
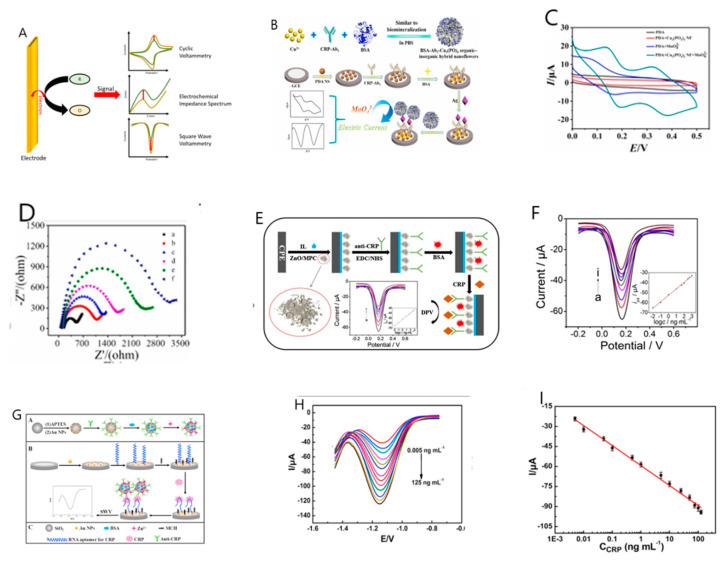
(**A**) Electrochemical sensor basic structure. (**B**) Illustration of the synthesis procedure of BSA-Ab2-Cu3(PO4)2 hybrid nanoflowers as signal labels, fabrication process of the electrochemical immunosensor. (**C**) Cyclic voltammetry (CV) of polydopamine nanospheres (PDANS) modified electrode (black line), BSA-Ab2-Cu_3_(PO4)_2_ nanoflowers on the modified electrode (red line), and sodium molybdate reaction with (green line), without BSA-Ab2-Cu_3_(PO4)_2_ nanoflowers on the modified electrode (blue line). (**D**) Electrochemical Impedance Spectrum (EIS) responses recorded at different modified electrodes in 5.0 mM [Fe(CN)_6_]^3−/4−^ and 0.1 M KCl. From curve a to curve f: bare GCE; PDANS/GCE; Ab1/PDA NS/GCE; BSA/Ab1/PDANS/GCE; CRP/BSA/Ab1/PDANS/GCE; BSA-Ab2-Cu_3_(PO4)_2_/CRP/BSA/Ab1/PDANS/GCE, respectively. (**E**) Schematic diagram of the processes to prepare C-reactive protein (CRP) immunosensor. (**F**) Differential Pulse Voltammetry (DPV) responses of BSA/anti-CRP/ZnO/MPC/IL-CPE to different CRP concentration (0.01, 0.1, 1, 10, 50, 100, 300, 500, 1000 ng·mL^−1^). Inset: linear relationship of current (Ip) vs. logarithm (logc). (**G**) Schematic diagram of the processes to prepare CRP aptasensor. (**H**) Square Wave Voltammetry (SWV) recorded of the aptasensor at different concentrations of CRP (0.005, 0.01, 0.05, 0.1, 0.5, 1.0, 5.0, 10.0, 25.0, 50.0, 75.0, 100.0, 125.0 ng mL^−1^) in 0.2 M HAc-NaAc (pH 5.5). (**I**) Calibration curve of the peak current vs the logarithmic concentration of CRP. (Reproduced with permission from [62,71,77] published by Elsevier 2019, 2017).

**Figure 3 sensors-21-03024-f003:**
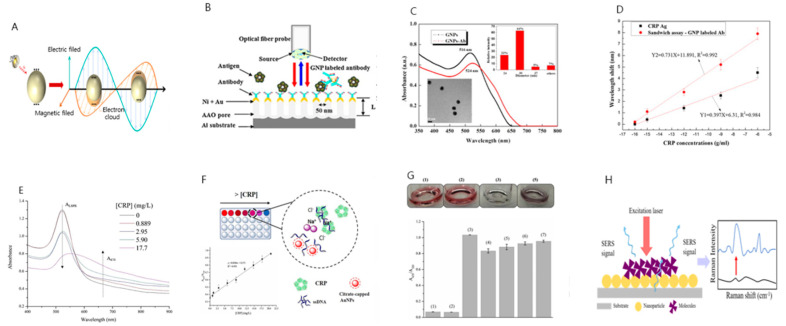
(**A**) Local surface plasmon resonance (LSPR) basic structure. (**B**) Schematic cross-sectional view showing the structure of an AAO chip fabricated for CRP antigen detection. (**C**) Absorption spectra of gold nanoparticle (GNP) and GNP-labeled CRP antibodies. (**D**) CRP antigen–antibody response (black), and linear regression of the resonant wavelength shift after GNP-labeled CRP secondary antibody reaction (red). (**E**) Overall schematic diagram of CRP detection method. (**F**) (1) AuNP colloid, (2) Bottom-ratio value of Apt-AuNPs Set it. (3) Add CRP 17.7 mg/L and then Au colloid, (4) CRP 17.7 mg/L Apt-AuNPs, (5) NaCl (60 μL, 0.2 M) after addition (4), (6) (4) CRP Same as 20.7 mg/L, (7) (5) same as CRP 20.7 mg/L; Photos of (1), (2), (3) and (5), (**G**) UV-vis absorbance spectra of top samples of colloids of Apt-AuNPs for CRP addition at various concentrations. (**H**) Surface-enhanced Raman spectroscopy (SERS) basic principle (Reproduced with permission from [105,108] published by Elsevier 2013, 2020).

**Table 1 sensors-21-03024-t001:** The biosensing techniques using electrical means of detection techniques discussed in this review.

Probe	Detection Method	Detection Range	LOD	Ref
Antibody	FET	0.01~1000 ng/mL	0.01 ng/mL	[37]
Antibody	FET	10~1000 ng/mL	10 ng/mL	[38]
Antibody	FET	1~200 ug/mL	-	[1]
Antibody	Interdigitated capacitor	25~800 ng/mL	-	[2]
Antibody	Interdigitated capacitor	25 pg/mL~25 ng/mL	32 pg/mL	[3]
RNA aptamer	Capacitance	100~500 pg/mL	-	[4]

**Table 2 sensors-21-03024-t002:** Summary of electrochemical biosensing techniques discussed in this review.

Probe	Detection Method	Detection Range	LOD	Ref
Antibody	SWV	5–1000 pg/mL	1.26 pg/mL	[62]
Antibody	DPV	0.01–1000 ng/mL	5.0 pg/mL	[71]
Aptamer	SWV	1 to 125 ng/mL	0.0017 ng/mL	[77]
Aptamer	EIS	1pM–100nM	0.349 pM (buffer) 3.55 fM (20% human serum)	[80]

**Table 3 sensors-21-03024-t003:** Summary of surface plasmon resonance (SPR) biosensing techniques.

Probe	Detection Method	Detection Range	LOD	Ref
Antibody	SPR	0.006–70 mg/L	0.009 mg/L	[93]
Aptamer	SPR	10 pM–100 nM	10 pM	[94]
Antibody	SPR	2–5 μg		[95]
Antibody	LSPR	100 ag/mL–1 fg/mL	100 ag/mL	[104]
Antibody	LSPR	0–3000 ng/mL	50 ng/mL	[105]
Antibody	SERS	0.1–0.01 ng/mL	0.1 ng/mL	[115]
Antibody	SERS	0.1 pg/mL–1 μg/mL	0.01 pg mL^−1^	[116]
Aptamer	colorimetric immunoassay	0.889–20.7 mg/L	1.2 μg/mL	[107]
Antibody	colorimetric immunoassay	10 ng/mL–5 μg/mL	100 ng mL^−1^	[108]

## Data Availability

Not Applicable.

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
