# Peer review of "Recent Advances in CRP Biosensor Based on Electrical, Electrochemical and Optical Methods"

_sensors, 2021, doi:10.3390/s21093024_

Round 1
Reviewer 1 Report
In this manuscript, the authors review three types of biosensors, based on the detection of the C-reactive protein (CRP), whose role as disease marker is of great importance in inflammatory/infectious process. Great emphasis is made on the use of various nanomaterials applied in the presented electrical, electrochemical and SPR biosensors architecture; due to the fact that they enhance the detection of target molecules.
The manuscript is well written and interesting; however, it lacks some critical involvement from the author’s side. Readers expect from review articles not only a summary of the most recent advances in the field, but also comments, comparisons, highlighting weak and strong points of the presented CRP biosensors.
The authors of chapter 3 should carefully revise the section for clarity. Some sentences are too long and the train of thoughts is sometimes difficult to follow. Also, the authors should differentiate the classical immunosensors from the aptamer-based immunosensors and add a short discussion at the end of the chapter about advantages and disadvantages between the two immunosensor types (for e.g. aptamers must be engineered to be specific and are therefore very expensive).
Some parts of the manuscript need more attention as follows:
- line 54 : “commercially CRP assays” – please clarify: do the authors mean commercially available kits for personal testing or well established clinical laboratory protocols
- delimitation and title of chapter 2 is missing
- line 98: “AlGaN/GaN layers” explanation of abbreviation is missing, as well as line 406 and more – please carefully check the manuscript
- lines 157-160: “In electrochemical biosensors….. [55]” – rephrasing is needed due lack of clarity
- lines 163-166: “Recently, the Tang …… hybrid nanoflowers (BSA-Ab2-Cu3(PO4)2)” – rephrasing is needed due lack of clarity
- the word “immune sensor” is repeatedly misspelled instead of correct “immunosensor”
- lines 211-215: – rephrasing is needed due lack of clarity
- lines 297, 332: “11-mercaptoundecanoic acid” is written differently. Once abbreviated (line 105), use abbreviation
- lines 428-441: there is too much emphasis on the impact of nanomaterials instead of discussing biosensor architecture, ease of reproducibility, reliability and selectivity, which are essential parameters for biosensors. Please also add a discussion on these parameters.
Author Response
Some parts of the manuscript need more attention as follows:
line 54 : “commercially CRP assays” – please clarify: do the authors mean commercially available kits for personal testing or well established clinical laboratory protocols
- The author agreed with the reviewer's opinion and deleted the “commercially CRP assays” and described in the manuscript how the CRP level test was conducted. Please chcek line 59~61 with the red color.
delimitation and title of chapter 2 is missing
- The author has confirmed the comments and revised the chapter 2 title. Please check line 74 with the red color.
line 98: “AlGaN/GaN layers” explanation of abbreviation is missing, as well as line 406 and more – please carefully check the manuscript
- AlGaN/GaN is a heterogeneous layer of Aluminum Gallium Nitride and Gallium Nitride, and is an element symbol, not an abbreviation. Therefore, the author did not feel the need to explain the abbreviation.
lines 157-160: “In electrochemical biosensors….. [55]” – rephrasing is needed due lack of clarity
- Authors agreed with reviewer's question. We modified it in the revised manuscript. Please check line 182-183 with the red color. ' The nanomaterial's crystal structure, surface ~ '.
lines 163-166: “Recently, the Tang …… hybrid nanoflowers (BSA-Ab2-Cu3(PO4)2)” – rephrasing is needed due lack of clarity
- Authors agreed with reviewer's question. We modified it in the revised manuscript. Please check line 186-190 with the red color. ' Recently, the Tang group published a study~ '.
the word “immune sensor” is repeatedly misspelled instead of correct “immunosensor”
- The authors have revised the relevant sentences in agreement with the reviewer's opinion.
lines 211-215: – rephrasing is needed due lack of clarity
- Authors agreed with reviewer's question. We modified it in the revised manuscript. Please check line 226-228 with the red color. ' For quantitative determination of various CRP concentrations (0.01~1000 ng/mL), the performance ~'.
lines 297, 332: “11-mercaptoundecanoic acid” is written differently. Once abbreviated (line 105), use abbreviation
- TAuthors agreed with reviewer's question. We modified it in the revised manuscript. Please check line 325,364 with the red color. ' The SPR biosensor used a solution of 11-MUA ~', ' A thiol group was subsequently formed on ~'.
lines 428-441: there is too much emphasis on the impact of nanomaterials instead of discussing biosensor architecture, ease of reproducibility, reliability and selectivity, which are essential parameters for biosensors. Please also add a discussion on these parameters.
- In agreement with the reviewer's opinion, the authors added a discussion of the biosensor.
Reviewer 2 Report
CRP assay is widely used as a representative marker of acute and chronic inflammatory diseases. This review describes the impact of nano-biohybrid materials on CRP biosensor design. The sensors include electrochemical, electrical and spectroscopic CRP biosensors. I do not think this review is an adequate summary of the existing literature. I recommend that the authors resubmit the manuscript with revisions. Specific comments 1.The authors do not seem to cite their own team's published papers in this area. I would like to see the results of the authors' team for the CRP assay and how their results compare to the existing results. 2.Section 2 is MISSING!!! 3.The authors detail many of the team's research efforts, but there is no connection between these works. Is a particular work a methodological improvement of a previous one? Or is it just a change of nanomaterials? Why does a change of nanomaterial work better? 4.There are many typos in the article content. 5.The authors describe several different sensors, but do not compare their advantages and disadvantages. 6.I'm not sure why the authors emphasize Eq1-3.Author Response
CRP assay is widely used as a representative marker of acute and chronic inflammatory diseases. This review describes the impact of nano-biohybrid materials on CRP biosensor design. The sensors include electrochemical, electrical and spectroscopic CRP biosensors. I do not think this review is an adequate summary of the existing literature. I recommend that the authors resubmit the manuscript with revisions. Specific comments
1.The authors do not seem to cite their own team's published papers in this area. I would like to see the results of the authors' team for the CRP assay and how their results compare to the existing results.
- Section 3 is listed in the contents and in Table 2.
2.Section 2 is MISSING!!!
- The author has confirmed the comments and revised the chapter 2 title. Please check line 74 with the red color
3.The authors detail many of the team's research efforts, but there is no connection between these works. Is a particular work a methodological improvement of a previous one? Or is it just a change of nanomaterials? Why does a change of nanomaterial work better?
-> The use of nanomaterial in the biosensor fabrication has several goals. 1) Enhance the detection sensitivity by introducing the nanomaterial. Especially, the electrical biosensor and electrochemical biosensor can be tuned with nanomaterial. We didn’t know which nanomaterial provides the best performance. That’s why the various nanomaterial can be applied to biosensor. 2) Various nanomaterial can be used to develop the new concept of detection mechanism including optical-based biosensor such as SERS or SPR. 3) Finally, some of nanomaterial facilitate the electron transfer between redox species and target that enhance the biosensor sensitivity or selectivity.
4.There are many typos in the article content
- In agreement with the reviewer's opinion, the authors corrected the typo entirely.
5.The authors describe several different sensors, but do not compare their advantages and disadvantages.
- Authors agreed with reviewer's question. We modified it in the revised manuscript. Please check line 141-148, 265-269, 297-299 with the red color.
6.I'm not sure why the authors emphasize Eq1-3.
- The authors agreed with the reviewer and excluded the equation from the manuscript
Reviewer 3 Report
This review article discusses the recent advances in CRP based biosensors that use three main transduction methods, namely electrical, electrochemical and optical. There are a fair amount of grammatical errors scattered throughout the article. It needs to be proofread to a publication standard before resubmission. I have tried to highlight a few but there are many more. The literature is well assessed; however, it is poorly organised. It requires major revision before publication.
Other comments are listed below;
- In the abstract the usage of the terms nanobiotechnology and nanohybrid material is very vague and confusing. I suggest rephrasing to be more accurate.
- Page 1 line 31-32. This sentence needs to be rewritten- Grammatically not right
- Line 66- What is an electrical detection sensor. Do they mean FET or chemiresistors? Need to be clearer. Maybe it can be changed to ‘Electrical means of detection’
- Line 70- Grammatical error- ‘and is has gained popularity’
- Line 70-71 What is a nano biosensor? A description in the introduction is needed
- After the introduction section. It jumps to Electrochemical section. Where is section 2? I assume it is the FET section. It needs to be clearly stated and organised properly.
- At the beginning of the electrochemical section, it will be advantageous to have a working principle diagram. What techniques are common? Why are they used? These should all be addressed before looking through the literature.
- Usage of the term ‘A well-designed nanomaterial’ doesn’t add any value. Please describe this.
- In the electrochemical biosensors section- Why are certain materials used? It will be better if they separate it according to bioreceptors used or electroanalytical techniques used. There is no flow the way it is currently written
- Again it will be better to have a working principle diagram of how optical biosensors work. I will have subsections here for the different types of optical transduction techniques so that it is clearer.
- They should have a paragraph in the future perspectives section with the shortcomings of current research and give suggestions and recommendations for future research. This is an important element for review articles. They have discussed the issues of toxicity of nanomaterials but more is needed based on techniques that were used for sensing.
Author Response
In the abstract the usage of the terms nanobiotechnology and nanohybrid material is very vague and confusing. I suggest rephrasing to be more accurate.
Page 1 line 31-32. This sentence needs to be rewritten- Grammatically not right
- In agreement with the reviewer's opinion, the author revised the relevant sentence. Please check line 33-39 with the red color
Line 66- What is an electrical detection sensor. Do they mean FET or chemiresistors? Need to be clearer. Maybe it can be changed to ‘Electrical means of detection’
- In agreement with the reviewer's opinion, the author changed the title of the chapter to "Field effect transistor based biosensors" and revised the manuscript. Please check line 73~81 with the red color.
Line 70- Grammatical error- ‘and is has gained popularity’
- In agreement with the reviewer's opinion, the author revised the relevant sentence. Please check line 70~72 with the red color.
Line 70-71 What is a nano biosensor? A description in the introduction is needed
- In agreement with the reviewer's opinion, the author revised the relevant sentence. Please check line 70~72 with the red color.
After the introduction section. It jumps to Electrochemical section. Where is section 2? I assume it is the FET section. It needs to be clearly stated and organised properly.
- The author has confirmed the comments and revised the chapter 2 title. Please check line 73 with the red color.
The At the beginning of the electrochemical section, it will be advantageous to have a working principle diagram. What techniques are common? Why are they used? These should all be addressed before looking through the literature.
- Authors agreed with reviewer's question. We modified it in the revised manuscript. Please check line 169-178 with the red color. ' Electrochemistry is the study of electricity ~'.
Usage of the term ‘A well-designed nanomaterial’ doesn’t add any value. Please describe this.
- Authors agreed with reviewer's question. We modified it in the revised manuscript. Please check line 180, 468 with the red color.
In the electrochemical biosensors section- Why are certain materials used? It will be better if they separate it according to bioreceptors used or electroanalytical techniques used. There is no flow the way it is currently written
- Authors agreed with reviewer's question. We modified it in the revised manuscript. The electrochemical analysis techniques are listed in lines 169-178 and nanomaterials are listed in lines 180-186.
Again it will be better to have a working principle diagram of how optical biosensors work. I will have subsections here for the different types of optical transduction techniques so that it is clearer.-
- According to the reviewer's opinion, the author added the principle of optical biosensor to Chapter 4. Please check line 297-305 with the red color. ' Optical biosensors utilize an electric field, which is ~'.
They should have a paragraph in the future perspectives section with the shortcomings of current research and give suggestions and recommendations for future research. This is an important element for review articles. They have discussed the issues of toxicity of nanomaterials but more is needed based on techniques that were used for sensing.
- Authors agreed with reviewer's question. We modified it in the revised manuscript. Please check line 461-463 with the red color. ' The biosensor reacts with targets to be measured ~'.
Reviewer 4 Report
The manuscript entitled “Recent Advances in CRP biosensor Based on Electrical/Electro-chemical/Optical Methods.” The manuscript is well written and a very comprehensive overview of CRP-based biosensors; however, it requires refinement to improve the manuscript's quality. Nonetheless, the manuscript can and should be further improved by taking the following aspects into consideration Main comments 1. The introduction needs to be extended a bit in more information about the use of Electrical/Electro-chemical/Optical Methods for biosensing. Adding some historical information like state-of-art, how many papers have been published so far (rough estimates), relevant technologies if there exists any should be an overview, and a clear motivation should be highlighted behind the need of such a review article for the community. 2. A very important point and was missed by the authors: what kind of materials show biosensing. Please justify this in the review. 3. In the last 20 years, there is no such improvement in selectivity for biosensors except miniaturization. Selectivity is still a challenge. How this sensor address overcoming these problems. 4. It is important to describe the sensing mechanism in each section. 5. It is important to mention some material such as metal oxides, carbon materials, 2D materials, which are frequently used in biosensing and their advantages and disadvantages: Cite some papers: Nano-Micro Lett. 12, 122 (2020), Journal of the Taiwan Institute of Chemical Engineers 116 (2020) 26-35, Sensors 2021, 21(6), 1939, Sensors 2021, 21(6), 2148, Journal of the Taiwan Institute of Chemical Engineers 118 (2021) 245-253. 6. Is it better to elaborate on the scope and limitations of the CRP biosensor in the introduction part? 7. The quality of some figures is inferior and needs to be enhanced. 8. Every section needs to be accompanied by adequate discussions about advantages and disadvantages and a clear connecting motivation to the next section. 9. Some discussions about sensing mechanisms need to be included in every section. 10. It is better to check and correct the font size of the x and y-axis of all figures in the manuscript. It should be the same. 11. The quality of some figures is inferior and needs to be enhanced. 12. The first paragraph contains trivial statements. The introduction should be reduced in length and have a focus on current analytical challenges. Essential related works can be cited.Author Response
The manuscript entitled “Recent Advances in CRP biosensor Based on Electrical/Electro-chemical/Optical Methods.” The manuscript is well written and a very comprehensive overview of CRP-based biosensors; however, it requires refinement to improve the manuscript's quality. Nonetheless, the manuscript can and should be further improved by taking the following aspects into consideration Main comments
- The introduction needs to be extended a bit in more information about the use of Electrical/Electro-chemical/Optical Methods for biosensing. Adding some historical information like state-of-art, how many papers have been published so far (rough estimates), relevant technologies if there exists any should be an overview, and a clear motivation should be highlighted behind the need of such a review article for the community.
- In agreement with the reviewer's opinion, the author revised the "Introduction" section of the manuscript to give it a clear motivation. Please check line 33-39, 58-72 with the red color.
- A very important point and was missed by the authors: what kind of materials show biosensing. Please justify this in the review.
- Antibodies or aptamers selectively bind to CRP, causing an antigen-antibody reaction. Therefore, the signal changes before and after the antigen-antibody reaction, and sensing is performed using this. The mechanism for this was described in each session.
- In the last 20 years, there is no such improvement in selectivity for biosensors except miniaturization. Selectivity is still a challenge. How this sensor address overcoming these problems.
-> Authors agreed with reviewer’s question. The enhancement of selectivity can’t control by introducing the nanomaterial. Selectivity related with dissociation constant between target molecule and antibody/aptamer. It can be tuned with antibody production or aptamer selection. The present reviewer focused on the improvement of sensitivity by introducing nanomaterial.
- It is important to describe the sensing mechanism in each section. –
- Authors agreed with reviewer's question. We modified it in the revised manuscript. Please check line 141-148, 169-178, 299-310 with the red color.
- It is important to mention some material such as metal oxides, carbon materials, 2D materials, which are frequently used in biosensing and their advantages and disadvantages: Cite some papers: Nano-Micro Lett. 12, 122 (2020), Journal of the Taiwan Institute of Chemical Engineers 116 (2020) 26-35, Sensors 2021, 21(6), 1939, Sensors 2021, 21(6), 2148, Journal of the Taiwan Institute of Chemical Engineers 118 (2021) 245-253.
- In agreement with the reviewer's opinion, the authors added the reference [34]..
- Is it better to elaborate on the scope and limitations of the CRP biosensor in the introduction part?
- The detection range and limits of CRP biosensors vary from study to study, so are shown in the table at the end of each section.
- The quality of some figures is inferior and needs to be enhanced.
- In agreement with the reviewer's opinion, the authors revised the figure.
- Every section needs to be accompanied by adequate discussions about advantages and disadvantages and a clear connecting motivation to the next section.
- Authors agreed with reviewer's question. We modified it in the revised manuscript. Please check line 141-148, 169-178 with the red color.
- Some discussions about sensing mechanisms need to be included in every section.
- Authors agreed with reviewer's question. We modified it in the revised manuscript. Please check line 141-148, 169-178, 299-310 with the red color.
- It is better to check and correct the font size of the x and y-axis of all figures in the manuscript. It should be the same.
- In agreement with the reviewer's opinion, the authors revised the figure.
- The quality of some figures is inferior and needs to be enhanced.
- In agreement with the reviewer's opinion, the authors revised the figure.
- The first paragraph contains trivial statements. The introduction should be reduced in length and have a focus on current analytical challenges. Essential related works can be cited.
- In agreement with the reviewer's opinion, the author revised the first paragraph to be related to the analysis task. Please check line 33-39 with the red color.
Round 2
Reviewer 1 Report
Manuscript was improved and can be published in current form.
Author Response
Thanks for reviewer's suggestion.
Reviewer 2 Report
The article has improved significantly after the revision. I think this review can be accepted for publication.
Author Response
Thank you for reviewer's response.
Reviewer 3 Report
The authors have made some changes to the manuscript, however the English and grammar is still very poor in the article. Some changes to improve the flow of the review was not considered such as including a section on bioreceptors. There are no working principle diagrams of Electrochemical and optical sensors. The future perspectives section is very weak and doesn’t suggest recommendations for the wider community. It is very hard to understand the meanings of sentences and I do not recommend publication at this current state. Furthermore, other than the specific changes asked to be made, no changes to the organisation of the literature review was considered.
Some specific corrections below.
I would rename the title to be 'Recent Advances in CRP biosensor Based on Electrical, Electrochemical and Optical Methods'
line 28 - Why is selectivity hyphenated in the abstract?
Line 37 - Grammatically not right- ‘To effectively cope with these risks requires rapid diagnosis and appropriate 37 treatment for the disease.’
Line 73 Typo- ‘Field effet transistor based biosensors’
Line 71- grammatical error- in this 70 paper reviews,
There are no references for lines 170 to 180.
Author Response
The authors have made some changes to the manuscript, however the English and grammar is still very poor in the article.
- Before we submit the manuscript, the manuscript corrected the English correction service. Please find the attached certificate of the manuscript.
Some changes to improve the flow of the review was not considered such as including a section on bioreceptors. There are no working principle diagrams of Electrochemical and optical sensors.
- Authors agreed with reviewer’s question. We added working principle of Electrochemical and optical sensors.
The future perspectives section is very weak and doesn’t suggest recommendations for the wider community.
- Authors agreed with reviewer’s question. We modified the future prospective.
It is very hard to understand the meanings of sentences and I do not recommend publication at this current state. Furthermore, other than the specific changes asked to be made, no changes to the organisation of the literature review was considered.
.There are no working principle diagrams of Electrochemical and optical sensors.
- Authors agreed with reviewer’s question. We added working principle of Electrochemical and optical sensors.
- Authors agreed with reviewer's question. We modified it in the revised manuscript. Please check figure 2(A), 3(A), 3(H) and line 174, 346-348,4 30-434 with the blue color. ' Figure 2 (A) shows a diagram of ~, Figure 3(A) shows the basic principle ~, Figure 3 (H) shows the basic principle of Surface-enhanced Raman spectroscopy (SERS)~'.
Some specific corrections below.
I would rename the title to be 'Recent Advances in CRP biosensor Based on Electrical, Electrochemical and Optical Methods'
- Authors revised the title based on reviewer's opinion.
line 28 - Why is selectivity hyphenated in the abstract?
- Authors agreed with the reviewer and removed the hyphen.
Line 37 - Grammatically not right- ‘To effectively cope with these risks requires rapid diagnosis and appropriate 37 treatment for the disease.’
- The grammatical error was corrected by removing the sentence from the manuscript. Pleas check line 34~39.
Line 73 Typo- ‘Field effet transistor based biosensors’
- It is corrected the typo in "effect" and marked it in blue.
“Field effect transistor based biosensors”
Line 71- grammatical error- in this 70 paper reviews,
- To correct grammatical errors, the author corrected the sentence as follows and marked it in blue.
“To shed light on the current progress of the CRP detection platform, we review the recent advances in CRP detection systems divided into three sections, including the latest research trends based on EC, SPR, SERS., etc.”
There are no references for lines 170 to 180.
- Authors agreed with the reviewer and added references. [57,58,59]

Reviewer 4 Report
All the comments addressed properly, the manuscript is ready for the publication
Author Response
Thank you for reviewer's valuable comment.